# Utilization of Whey for Red Pigment Production by *Monascus purpureus* in Submerged Fermentation

**Dilara Mehri [1], N. Altinay Perendeci [2] and Yekta Goksungur [1],***

[1] Department of Food Engineering, Ege University, İzmir 35100, Turkey; dilara.mehrii@gmail.com
[2] Environmental Engineering Department, Akdeniz University, Antalya 07058, Turkey; aperendeci@akdeniz.edu.tr
* Correspondence: yekta.goksungur@ege.edu.tr; Tel.: +90-232-311-3027

**Abstract:** Various biotechnological approaches have been employed to convert food waste into value-added bioproducts through fermentation processes. Whey, a major waste generated by dairy industries, is considered an important environmental pollutant due to its massive production and high organic content. The purpose of this study is to investigate the effect of different fermentation parameters in simultaneous hydrolysis and fermentation (SHF) of whey for pigment production with *Monascus purpureus*. The submerged culture fermentation parameters optimized were type and pretreatment of whey, pH, inoculation ratio, substrate concentration and monosodium glutamate (MSG) concentration. Demineralized (DM), deproteinized (DP), and raw whey (W) powders were used as a substrate for pigment production by simultaneous hydrolysis and fermentation (SHF). The maximum red pigment production was obtained as 38.4 $UA_{510\,nm}$ (absorbance units) at the optimized condition of SHF. Optimal conditions of SHF were 2% (*v/v*) inoculation ratio, 75 g/L of lactose as carbon source, 25 g/L of MSG as nitrogen source, and fermentation medium pH of 7.0. The specific growth rate of *M. purpureus* on whey and the maximum pigment production yield values were 0.023 $h^{-1}$ and 4.55 $UAd^{-1}$, respectively. This study is the first in the literature to show that DM whey is a sustainable substrate in the fermentation process of the *M. purpureus* red pigment.

**Keywords:** microbial red pigment; *Monascus purpureus*; simultaneous hydrolysis and fermentation; sustainability; whey

## 1. Introduction

Color introduces data to the class of products and particular brands, creating effective visual recommendations to show attractiveness, tastes, distinction, and novelty in packages. Food producers have preferred to use natural pigments instead of artificial pigments in response to the developing consumer perception that natural pigments are safer. Plant-originated pigments have some disadvantages, such as low solubility in water, being unstable against heat and light [1,2], and high prices along with the required large agricultural areas. However, microbial pigments have many advantages such as being independent of seasonal changes, use of low-cost raw materials, being biodegradable, having different color tones depending on the conditions of culture and higher efficiency compared to chemical synthesis [1,3]. Thus, microbial pigments identified as bio-pigments are the safer preference for the food industry. The production of microbial pigments has been gradually increasing since health concerns such as carcinogenicity and teratogenicity are related more to the synthetic pigments [4]. Furthermore, environmental policies are other main drivers for the diffusion of biopigments in the market.

Among the various pigment-producing organisms, *Monascus purpureus,* is a fungus that is isolated from red-fermented rice in Indonesia [5]. This filamentous fungus has been extensively preferred in the production of fermented foods in Asian countries, southern China, Japan, and Southeastern Asia [6]. *Monascus* pigments (MPs) as secondary metabolites have many therapeutic effects such as antioxidant, anti-inflammatory, antimicrobial, anticarcinogenic

and antimutagenic [7]. The main color pigments produced by *Monascus* spp. in polyketone structure are red, orange and yellow. The red ones of those six major pigments are called rubropunctamine ($C_{21}H_{26}NO_4$) and monascorubramine ($C_{23}H_{27}NO_4$); orange ones are called rubropunctatin ($C_{21}H_{22}O_5$) and monascorubrin ($C_{23}H_{26}O_5$) and yellow pigments are called as monascin ($C_{21}H_{26}O_5$) and ankaflavin ($C_{23}H_{30}O_5$) [8,9].

*Monascus* pigments are used as food additives, color thickener, or nitrite substitute in different types of foods (red wine, tofu, surimi, sausage, ham, different sauces, noodle products, ready meals, meat products, etc.) especially in East Asian countries. There are also application areas in the dairy, textile and cosmetics industries [1,4,10–12].

The wastes generated by the food industry cause serious environmental pollution and global warming. The major criteria of sustainable industrial production is the recovery and reuse of these wastes as a resource within the cycle of circular economy [13]. Hence, researchers in recent years have concentrated on studies about the use of food industry wastes in the production of high value biotechnological products. The utilization of wastes containing carbon and nitrogen in the bioprocesses is important in terms of reducing environmental pollution and also building low cost, robust and sustainable production schemes.

Various food industry wastes were used to produce *Monascus* pigments in the literature. These are; hydrolyzed rice straw [14], waste beer [15], brewer's spent grain [9], orange peels [16], chicken feather [17], sugarcane bagasse [18,19] bakery wastes [20], rice water based medium [21], sweet potato [22], corn cob [23,24], potato powder [1], the grape pulp [10], corn step liquor [25], Jack fruit wastes [26], wheat [5] and prickly pear juice [27].

Whey, which is a by product of cheese, casein and yogurt manufacturing, is considered as waste of the dairy industry [28]. When processed further, whey becomes a high value by product that is used as substrate in microbial fermentations utilizing lactose. However, it is defined as an ecologically harmful and most polluting waste when released directly into water receiving bodies [29]. With the rapid industrialization observed in the last century [30] and the growth rate of milk production (around 2.8% per annum), dairy processing is usually considered as the largest industrial food wastewater source, especially in Europe [31,32]. Depending on cheese type and production method, 150–200 kg of cheese is produced from a ton of milk, while 800–850 kg of whey is generated. Approximately 180–190 million tons of whey are produced annually in the world. It is a great threat to the environment [33] due to its very high biological oxygen demands (BOD > 35,000 ppm) and chemical oxygen demands (COD > 60,000 ppm) along with its low pH [34]. For a long time in the 20th century, the industry worked on an inexpensive removal strategy for whey, which included release into waterways, the ocean, municipal sewage treatment works, and/or onto fields. Today, these disposal methods are prevented by strict environmental regulations. Treatment and/or recovery of whey for its use in the production of value-added products have become a major concern. Since whey contains 4.5–5.0% lactose, 0.6–0.8% protein, 0.4–0.5% lipid, vitamins and minerals, there are many studies on its use as substrate in bioprocesses [34–36]. Since some microorganisms can not utilize lactose in whey as the carbon source, some bioprocesses require the hydrolysis of lactose into its monomers, glucose and galactose by the enzyme β-galactosidase.

The aim of this study is to investigate whey as an alternative low-cost sustainable substrate in the fermentation of *Monascus purpureus* CMU 001 strain to produce red color pigment for the food industry. Raw, demineralized, deproteinized whey as a substrate, fermentation pH, initial lactose concentration, monosodium glutamate (MSG) concentration as the nitrogen source, inoculation ratio, mycelial development and pigment synthesis kinetics of *Monascus purpureus* were studied. In contrast to the production of red pigments from different waste resources as a fermentation substrate, to the best of our knowledge no previous study in the literature has systematically investigated the simultaneous hydrolysis and fermentation of whey for the production of red pigment by *Monascus purpureus* from the point of sustainable resource recovery.

## 2. Materials and Methods

### 2.1. Microorganisms and Culture Media

The microorganism used in this study was the *Monascus purpureus* CMU 001 kindly provided by the Department of Biology, Chiang Mai University of Thailand. Our previous work [37] has showed that this strain did not produce the mycotoxin citrinin, which is a secondary metabolite of *Monascus* species. The strain was kept on potato dextrose agar (PDA; pH:5.6 ± 0.2, Merck, Darmstadt, Germany) at 4 °C and sub-cultured every 4 weeks to fresh PDA slants incubated at 30 °C for 7 days.

Raw whey powder, demineralized whey powder and deproteinized whey powder were supplied by Enka Süt ve Gıda Mamulleri Sanayi ve Ticaret A.S. located in Konya city, Turkey. All the whey samples were sweet whey-based samples as declared by the supplier. β-galactosidase (Saphera 2600L) enzyme was kindly donated by Novozymes A/S (Bagsvaerd, Denmark) for the hydrolysis of lactose in whey. Saphera 2600L is a bacterial β-galactosidase enzyme from *Bacillus licheniformis* with a specific activity of 2600 LAU-B/g (enzyme activity was stated as β-galactosidase that hydrolyzes terminal non-reducing β-D-galactosides releasing beta-D-galactose residues). Whey powder was diluted with distilled water to contain 50 g/L lactose concentration (unless otherwise stated) for fermentation experiments.

### 2.2. Preparation of Inoculum and Fermentation Medium

The inoculum was prepared by collecting spores from the surface of the PDA dishes under aseptic conditions with 10 mL of sterile distilled water. The spore suspension was used to prepare the inoculum culture containing $1.25 \times 10^6$ spores/mL. The spore concentration was estimated using a Neubauer chamber (Marienfeld-Superior, Lauda-Königshofen, Germany). The inoculation medium (50 mL in 250 mL Erlenmeyer flasks) was prepared by reconstituting whey powder in water to contain lactose at 50 g/L concentration and pH 6.0. Since *Monascus purpureus* can not utilize lactose as the carbon source, β-galactosidase enzyme (Saphera 2600L) was added to the fermentation medium at a rate of 0.1% (*v/v*) (simultaneous hydrolysis and fermentation, SHF). The fermentation for inoculum preparation was performed in a shaking incubator operated at 200 rpm, 30 °C for 4 days.

Fifty milliliters of demineralized whey medium (pH 6.0, unless otherwise stated) containing 50 g/L lactose was the substrate in SHF. The pH of the whey medium was adjusted to pH 6.0 by using 1 N potassium hydroxide and sterilized at 121 °C for 15 min. The inoculation ratio was 2% (*v/v*) and the SHF's were carried out in a shaking incubator at 200 rpm, 30 °C for 8 days. Inoculation and fermentation media were supplemented with 5 g/L of MSG (MSG; Sigma Aldrich, St Louis, MO, USA) as the nitrogen source. Samples were collected at equal time intervals for the determination of biomass growth rate and red pigment synthesis.

Prehydrolysis and a separate fermentation (PHSF) method was also used for the production of the red pigment. In this method, the prehydrolysis of lactose in demineralised whey medium was done at 50 °C (pH 6.0) for 8 h enzymatically before fermentation with *M. purpureus*.

### 2.3. The Effect of Whey Type on Pigment Production

Whey powder (W), demineralized whey powder (DM), deproteinized whey powder (DP), treated deproteinized raw whey powder (DPW), and treated deproteinized demineralized whey powder (DPDM) were used as fermentation substrate. Initial concentration of lactose was 50 g/L for each fermentation. DM and W samples were subjected to deproteinization to obtain DPW and DPDM. The process of deproteinization was adapted from Roukas et al., [38]; the pH of W and DM were adjusted to 5.0 and protein precipitation was induced by heating W and DM solutions at 90 °C for 20 min. The solutions were kept at room temperature for 1 h and precipitated proteins were removed by centrifugation (Hettich Universal 320 R, Tuttlingen, Germany) at 6000× *g* for 15 min.

## 2.4. Optimization of Fermentation Parameters

The experiments were conducted to find optimum fermentation conditions and to determine the effect of fermentation parameters on red pigment production. Different initial pH values (5.0, 6.0, 7.0, 8.0, 9.0), initial lactose concentrations (25, 50, 75, 100 g/L), MSG concentrations (2.5, 5.0, 7.5, 10.0, 12.5, 15.0, 20.0, 25.0, 30.0 g/L) as nitrogen source, and inoculation ratios of *Monascus* (1, 2, 3, 4, 5, 6%; (*v/v*)) were investigated for the pigment production. The pigment production kinetics and mycelial development were examined at the optimal conditions of fermentation determined from the experiments.

## 2.5. Analytical Methods

The biomass was measured gravimetrically at the end of fermentation by filtering the mycelia through Whatman No: 1 filter paper, washing three times with distilled water and drying at 65 °C to constant weight in an oven (Memmert, UN 55, Schwabach, Germany). The biomass concentration was expressed as mycelial dry weight per unit volume of culture medium [39].

The supernatant obtained by centrifugation of the fermentation medium in the centrifuge (Hettich Universal 320 R, Tuttlingen, Germany) at 5400 rpm/20 min was filtered with Whatman No. 1 filter paper. The filtrate obtained was used for the quantification of the synthesized red pigments using a spectrophotometer (Thermo Scientific, Genesys 10S UV-VIS, Paisley, UK) at 510 nm and the readings were given in absorbance units ($UA_{510}$) [40]. Sterile fermentation medium was used as the blank sample since the medium itself may have some absorbance at 510 nm wavelength. The pH values were determined with the pH-meter (WTW Inolab 7110, Weilheim, Germany). Lactose concentration in the medium was determined spectrophotometrically at 540 nm by the phenol-sulfuric acid method [41]. Total nitrogen content was determined by the Kjeldahl method [42] and the protein content of demineralized whey was calculated by multiplying the nitrogen content by the factor 6.25 in order to convert the nitrogen content into protein content. Moisture content of demineralized whey was determined by drying the sample to constant weight at 80 °C. The ash content of demineralized whey was determined by burning the sample for 12–18 h at 550 °C in a muffle furnace followed by cooling to room temperature and weighing.

Each experiment was performed in triplicate and the results were expressed as the mean ± standard deviation. The experiments were carried out in two repetitions and the analyses were carried out in three parallel samples. The results were expressed as the mean ± standard deviation. All statistical analyses were performed using the software IBM SPSS (v. 22). All data obtained were analyzed by one way analysis of variance, and tests of significant differences were determined by using Duncan's test at $p < 0.05$. In all the figures, mean values for all the factors given in x-axis that are not followed by the same letter (a–g) are significantly different ($p < 0.05$).

## 2.6. Cost Analysis

In this study, red pigment production cost was determined. The production cost includes the handling of raw materials (whey), chemicals (MSG and enzyme), electricity for fermentation, downstream processes (autoclaving, centrifuging, and drying) and the cost of water. Energy consumption values for fermentation and downstream processes were measured by a TT-Technic PM 001 plug power meter device. Prices of whey, chemicals, electricity, and water were obtained from the suppliers for cost analysis.

## 3. Results and Discussion

### 3.1. The Effect of Whey Type as Fermentation Substrate on Pigment Production

Raw whey powder (W), demineralized whey powder (DM), deproteinized whey powder (DP), treated deproteinized raw whey powder (DPW) and treated deproteinized demineralized whey powder (DPDM) were prepared to investigate the effect of whey type on pigment production by *Monascus purpureus*. The simultaneous hydrolysis and

fermentation (SHF) method was used in the fermentation experiments. The chemical composition of whey type has an important role in red pigment production since different types of whey have different protein and mineral contents [34].

As seen from Figure 1, the highest pigment concentration (20.8 UA$_{510\,nm}$) was produced from demineralized whey (DM) medium. Similar pigment yield value of 20.2 UA$_{510\,nm}$ was obtained with deproteinized and demineralized whey powder medium (DPDM). No significant difference ($p > 0.05$) in pigment yields were observed with DM and DPDM media. The lowest pigment yield was obtained from the DP medium (5.3 UA$_{510\,nm}$). While DP and DPW have yielded very low red pigment production values, DM and DPDM produced higher values. Lower pigment synthesis values observed in DP and PDW samples might have originated from the inhibitory effect of high concentration of cations (Na$^+$, NH$_4^+$, K$^+$, Mg$^{+2}$, and Ca$^{+2}$) and anions (Cl$^-$, SO$_4^{-2}$, PO$_4^{-3}$, and citrate) present in whey medium [43,44]. Since the demineralization process decreased monovalent, divalent cation and anion levels in whey, pigment production values obtained from DM and DPDM are satisfactory compared to the literature [45]. Raw whey powder (W) yielded 12.1 UA$_{510\,nm}$ pigment production value which was slightly higher than the values of DP and DPW as a substrate of pigment synthesis. The biomass concentration values observed in all types of whey were very close to each other and no significant correlation was observed between pigment and biomass production values (Figure 1). This indicates that biomass can reproduce in all types of whey, but the conditions for pigment production can be achieved at low anion and cation concentrations.

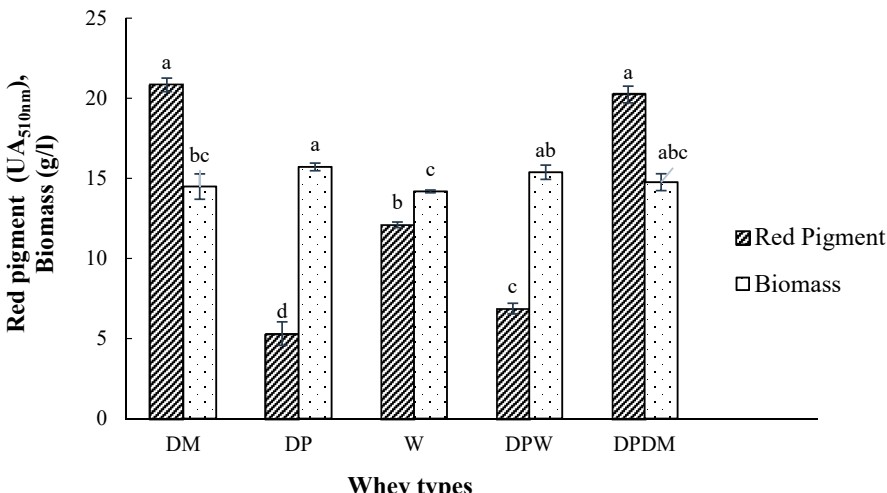

**Figure 1.** The effect of whey types on the red pigment and biomass production by *M. purpureus* CMU 001 (Fermentation conditions: 30 °C, 8 days, 200 rpm, pH 6.0, 5 g/L monosodium glutamate (MSG), 50 g/L lactose).

Demineralised whey powder with the highest pigment yield had the following composition (%): total sugar 70.46 ± 0.24, protein 5.32 ± 0.05, ash 3.32 ± 0.36, total nitrogen 0.86 ± 0.01, moisture 2.72 ± 0.3. As long as the lactose present in whey is enzymatically hydrolyzed, DM whey with its high lactose sugar content (70.5%) is a suitable substrate for *M. purpureus* in the synthesis of the red pigment.

### 3.2. Effects of Different Fermentation Methods on Pigment Production

Raw demineralised whey powder (DM) containing lactose as the carbon source in unhydrolyzed form was used for pigment production to determine the effect of lactose on growth and pigment production by *M. purpureus*. As seen in Figure 2, lactose found in DM was not utilized efficiently by *M. purpureus* for growth and pigment production. Other researcher also pointed out the difficulties of using lactose as the carbon source for the production of biopigments [45–48]. Therefore, it is necessary to hydrolyze lactose with β-galactosidase enzyme to its monomers of glucose and galactose for its proper utilization by *M. purpureus* as an energy source. In this study, two fermentation methods were

compared for lactose hydrolysis; simultaneous lactose hydrolysis and fermentation (SHF) and enzymatic pre-hydrolysis and separate fermentation (PHSF) for pigment production by *M. purpureus*. Demineralised whey powder (DM) containing 50 g/L lactose was used as the substrate in the fermentation experiments. As shown in Figure 2, the highest pigment production value (21.3 $UA_{510\ nm}$) was obtained with the SHF application. When lactose was hydrolyzed by the enzyme prior to fermentation (PHSF), lower red pigment synthesis (16.8 $UA_{510\ nm}$) was obtained compared to SHF. Similar biomass yields of 16.3 and 14.9 g/L were obtained for PHSF and SHF, respectively. Hence, SHF yielded higher pigment production values. Da Costa and Vendruscolo [45], determined pigment production by *Monascus ruber* CCT 3802 in the presence of several carbon sources such as glucose (20 g/L), lactose (20 g/L), and hydrolyzed lactose (20 g/L). Parallel to the results of our study, they found that the average production of yellow and orange pigments were higher in the hydrolyzed lactose medium by *M. ruber* compared to lactose.

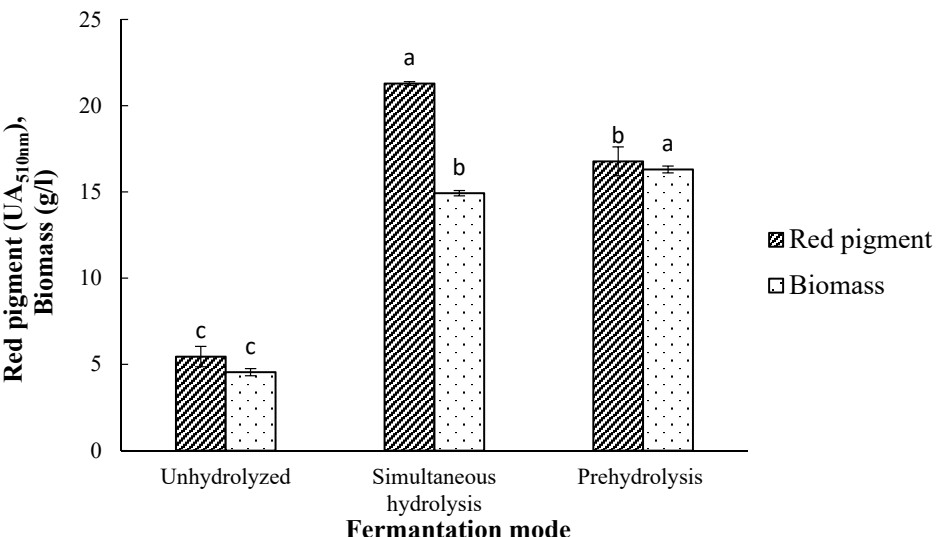

**Figure 2.** The effect of fermentation mode on the red pigment and biomass production by *M. purpureus* CMU 001 (Fermentation conditions: 30 °C, 8 days, 200 rpm, pH 6.0, 5 g/L MSG, 50 g/L lactose).

### 3.3. Effect of Initial pH on Pigment Production

The pH of fermentation medium is an important factor in red pigment synthesis by *Monascus* species since high pH values and the existence of suitable nitrogen source leads to the chemical modification of orange pigments changing into extracellular and water-soluble red pigments [49]. The effect of initial pH of fermentation medium was studied for pH range 5.0–9.0 using demineralised whey powder containing 50 g/L of lactose as the substrate. SHF (30 °C/8 days) was performed in 250 mL Erlenmayer flasks containing 50 mL of fermentation medium. As seen in Figure 3, the highest pigment synthesis of 25.3 $UA_{510\ nm}$ was observed at the initial pH value of 7.0. A low amount of pigment formation was observed at high and low pH values. In the fermentation medium with an initial pH of 9.0, growth and pigment synthesis were found to be the lowest (4.4 $UA_{510\ nm}$). Other researchers have also reported that pH of fermentation medium was highly important for red pigment synthesis by *M. purpureus* [48,50]. Orozco and Kilikian [50] investigated the synthesis of pigments by *Monascus purpureus* and they obtained 11.3 U of red pigments at pH 8.5. They stated that high pH medium facilitates the transfer of intracellular pigments to the fermentation medium. Parallel to the findings of our study, Mukherjee and Singh [39] stated that *M. purpureus* produced more red pigments at pH values of 6.0–8.0. Lee et al. [51] also indicated that the pH range of 5.5–8.5 was suitable for red pigment production, whereas pH values higher than 8.5 and lower than 5.5 led to a decrease in red pigment production. Prapajati et al. [48] achieved maximum *Monascus* pigment synthesis at pH 8.5. It has been reported that water-soluble extracellular

red pigment production increased at high pH values and at high MSG concentrations and the transition of pigments from the cell to fermentation medium was restricted at low pH values [49].

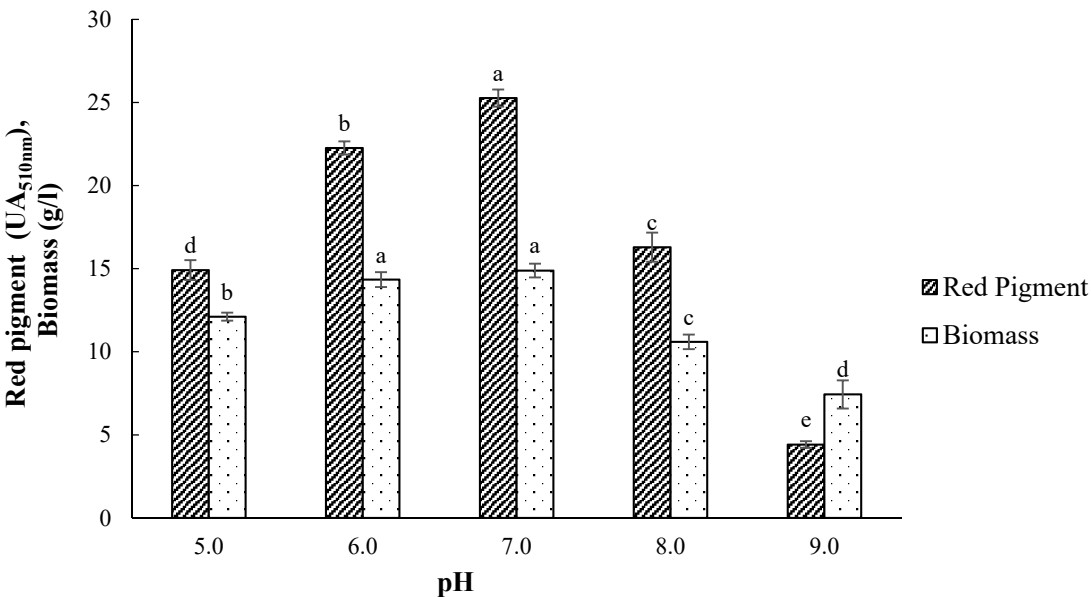

**Figure 3.** The effect of initial pH on the red pigment and biomass production by *M. purpureus* CMU 001 (Fermentation conditions: 30 °C, 8 days, 200 rpm, 5 g/L MSG, 50 g/L lactose).

### 3.4. Effect of Initial Lactose Concentration on Pigment Production

The effect of lactose concentration on pigment production by *M. purpureus* CMU 001 was investigated with DM whey containing different initial lactose concentrations as 25, 50, 75, and 100 g/L by SHF at 30 °C for 8 days. The initial pH of the fermentation medium was 7.0. The highest pigment density was determined as 27.7 $UA_{510 nm}$ in the fermentation medium containing 75 g/L of lactose (Figure 4). Red pigment concentrations of 10.2, 25.4, and 25.3 $UA_{510 nm}$ were found for the fermentation media containing initial lactose concentrations of 25, 50, and 100 g/L, respectively. Chen and Johns [52] reported that high glucose concentration of 50 g/L led to low biomass, pigment synthesis and ethanol production, while high maltose concentration of 50 g /L increased red pigment production by three fold compared to the same concentration of glucose. Liu et al., [53] studied pigment production by *M. purpureus* M183 and found the optimum glucose concentration as 80 g/L in terms of maximum efficiency, pigment yield and cost efficiency for industrial applications. Parallel to our findings, they stated that the initial substrate concentration has a negligible effect on biomass (dry cell weight). The high glucose concentration in the fermentation medium can be an advantage for mycelium growth, but as fermentation progresses, the fermentation medium becomes more acidic and this can lead to low pigment yields [54].

### 3.5. Effect of Monosodium Glutamate (MSG) Concentration on Pigment Production

The previous studies showed that *M. purpureus* had higher red pigment production efficiency in the presence of MSG than other nitrogen sources [9,15]. Pigments produced by *Monascus* species are usually intracellular and insoluble in water. These pigments turn into extracellular and water-soluble red pigments as a result of a non-enzymatic reactions in the presence of MSG at neutral pH values. MSG replaces the ammonia in the orange pigment to produce the red pigment derivatives [51,55–57]. To investigate the effect of MSG concentration on red pigment production, SHF was performed with DM whey containing 75 g/L of lactose at pH 7.0 and 30 °C temperature during 8 days. As seen in Figure 5, MSG concentration had a significant effect on red pigment synthesis

by *M. purpureus*. The pigment production increased steadily and linearly up to 25 g/L of MSG concentration resulting in a maximum pigment synthesis of 45.7 $UA_{510\ nm}$ and then declined at higher MSG concentrations. Atalay et al. [15] tested different nitrogen sources (monosodium glutamate, malt sprouts, corn steep liquor, peptone, urea, ammonium sulfate and yeast extract) for *Monascus* pigment production using residual beer as the fermentation medium and found that fermentation medium containing 7.5 g/L of MSG yielded the highest red pigment production of 18.5 $UA_{510\ nm}$. Sharmila et al. [1] obtained the maximum pigment synthesis by using 6 g/L of MSG as the nitrogen source. Babitha et al. [26] stated that Jackfruit seeds could not produce water-soluble pigments without using additional nitrogen sources. Zhang et al. [57] determined that glutamate and glycine was the most suitable source for growth of *Monascus*. Silbir and Göksungur [9] and Lee et al. [51] also used a submerged culture technique to determine the effect of various nitrogen sources on red pigment synthesis by *M. purpureus* and obtained maximum pigment production when MSG was used. Lee et al. [51] stated that increased MSG concentrations decreased pigment production while increasing biomass concentrations. Hamano and Kilikian [25] stated that the highest pigment production (20.7 U) was obtained in a fermentation medium containing 7.6 g/L of MSG concentration. Our results showed that the optimal MSG concentration found for DM is different and higher than the concentration values found for other substrates in the literature for the production of red pigment by *M. purpureus*. Demineralized whey medium might contain low concentration of nitrogen as stated before. When higher MSG concentrations are used to overcome this problem, *Monascus purpureus* gave higher pigment production values.

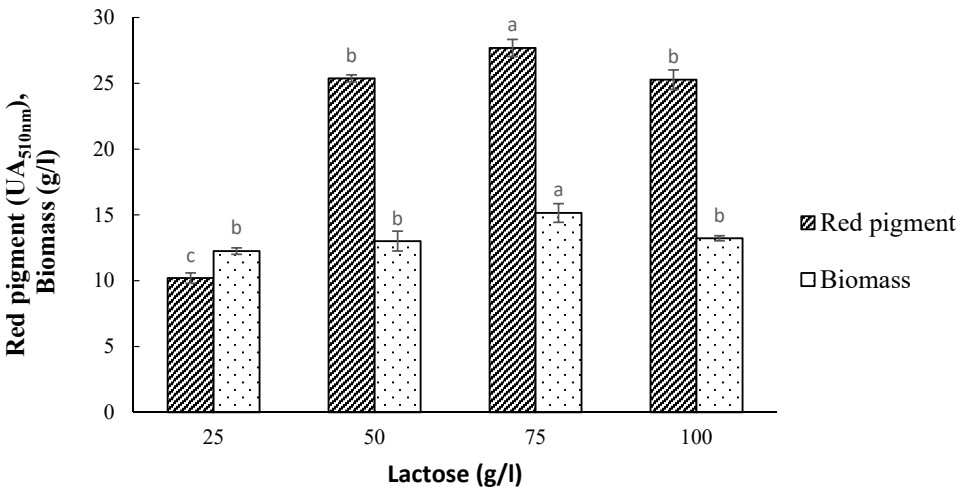

**Figure 4.** The effect of initial lactose concentration on red pigment and biomass production by *M. purpureus* CMU 001 (Fermentation conditions: 30 °C, 8 days, 200 rpm, pH 7.0, 5 g/L MSG).

*3.6. Effect of Inoculation Ratio on Pigment Production*

To investigate the effect of inoculation ratio on red pigment production, DM medium was inoculated with 1, 2, 3, 4, 5, and 6% (*v/v*) spore suspension solutions. SHF was performed at 30 °C for 8 days (pH 7.0) with initial lactose and MSG concentrations of 75 g/L and 5 g/L, respectively (Figure 6). Although maximum pigment synthesis was obtained at an MSG concentration of 25 g/L, an MSG concentration of 5 g/L was used in the fermentation experiments due to economical considerations and feasibility of the process. It was observed that the highest pigment production of 38.4 $UA_{510\ nm}$ was obtained when DM fermentation medium was inoculated with 5% of spore suspension. The results of this study showed that different inoculum ratios influenced the synthesis of the red pigment by *M. purpureus*, while 2 to 5% inoculum ratios had very little effect on the biomass concentration. Babitha et al. [26] studied pigment production by *M. purpureus* from jackfruit seed using solid-state fermentation and reported that low inoculum concentration produced insufficient biomass while high inoculum concentrations led to excessive

biomass formation depleting the nutrients in the fermentation medium that are necessary for the product formation. Atalay et al. [15] studied pigment production from residual beer using *M. purpureus* and obtained the highest pigment production of 18.5 $UA_{510\,nm}$ with the fermentation medium inoculated with 2% (*v/v*) of culture medium. Silbir and Goksungur [9] also studied pigment production from brewer's spent grain in submerged fermentation system and found that 2% (*v/v*) spore suspension yielded the highest pigment production of 22.3 $UA_{500\,nm}$.

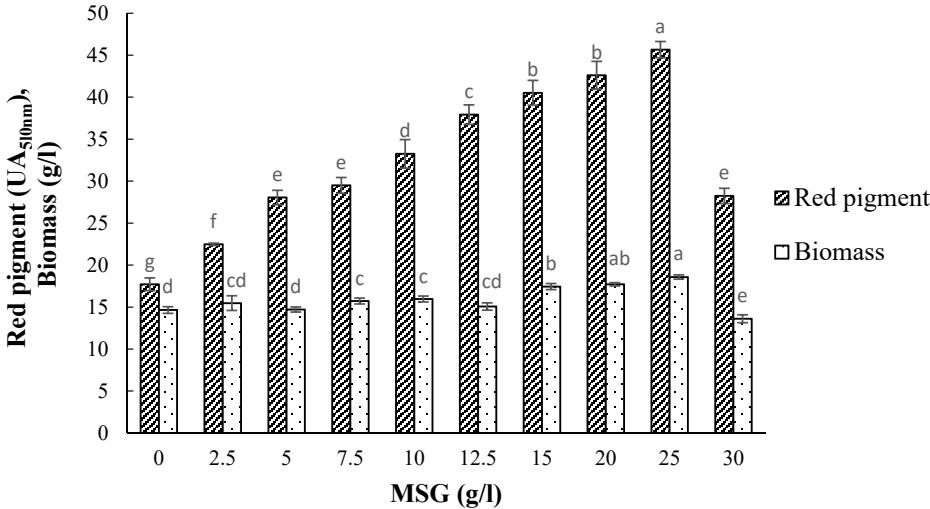

**Figure 5.** The effect of monosodium glutamate (MSG) concentration on the red pigment and biomass production by *M. purpureus* CMU 001 (Fermentation conditions: 30 °C, 8 days, 200 rpm, pH 7.0, 75 g/L lactose).

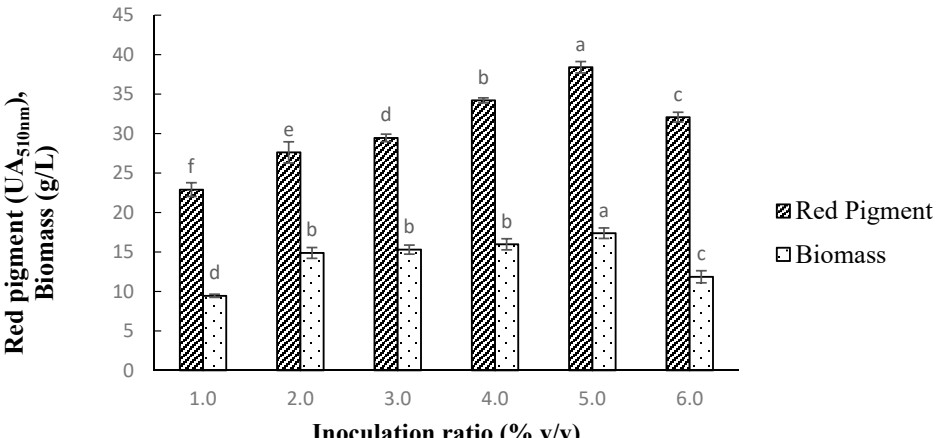

**Figure 6.** The effect of inoculation ratio on the red pigment and biomass production by *M. purpureus* CMU 001 (Fermentation conditions: 30 °C, 8 days, 200 rpm, pH 7.0, 5 g/L MSG, 75 g/L lactose.

### 3.7. Kinetics of Red Pigment Production by M. purpureus

The mycelial development and pigment synthesis kinetics of *M. purpureus* were investigated by SHF under the optimized conditions (demineralized whey diluted to contain 75 g/L lactose, 5 g/L MSG concentrations, pH 7.0, 30 °C, 200 rpm, and 8 days). As shown in Figure 7, red pigment production started at the beginning of the exponential growth phase and reached a maximum value of 37 $UA_{510\,nm}$ on the 8th days of fermentation. The red pigment concentration decreased after the 8th day of fermentation most probably due to the substrate limitation, possible chemical decomposition of the pigment, conversion to other products or oxidation by the microorganisms. The decrease in total sugar concentration during fermentation proved that hydrolyzed lactose was used by *M. purpureus* for growth

and pigment synthesis. The pH value dropped during the first 3 days of fermentation and increased afterwards to 8.0 at the end of the fermentation. The pH increase in the last stage of fermentation was probably due to *M. purpureus* producing ammonia as a result of the deamination of amino acids. Dry biomass weight steadily increased during the 8 days of fermentation period and then decreased.

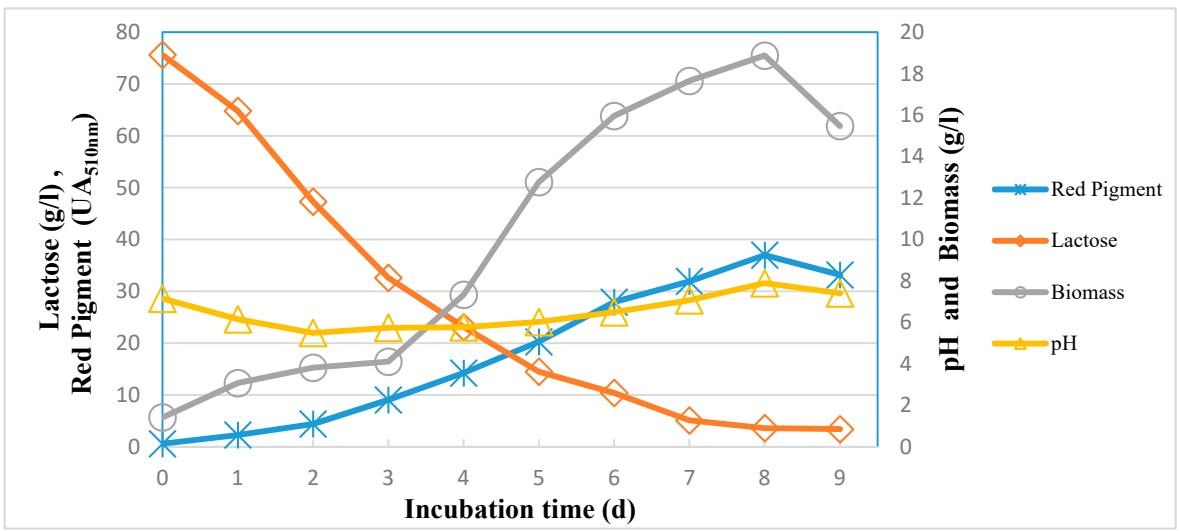

**Figure 7.** Simultaneous hydrolysis and fermentation (SHF) profiles of mycelial growth of *M. purpureus* CMU 001, red pigment synthesis, lactose consumption, and pH under the optimum conditions (Fermentation conditions: 30 °C, 8 days, 200 rpm, pH 7.0, 5 g/L MSG, 75 g/L lactose). The standard deviation of each experimental point ranged from 1.5 to 4.8.

Maximum pigment productivity and specific growth rate of *M. purpureus* were determined as 4.55 UAh$^{-1}$ and 0.023 h$^{-1}$, respectively for SHF of DM whey at the optimized conditions in this study. Da Costa and Vendrusculo [45] investigated the use of hydrolyzed lactose, glucose, and lactose as a substrate for pigment production. While productivity values were calculated as 0.059 AU$_{510}$ h$^{-1}$, 0.072 AU$_{510}$ h$^{-1}$, and 0.032 AU$_{510}$ h$^{-1}$, specific growth rate values were determined as 0.031 h$^{-1}$, 0.042 h$^{-1}$, and 0.017 h$^{-1}$ for hydrolyzed lactose, glucose, and lactose, respectively. Atalay et al. [15] produced 18.5 UA$_{510}$ red pigment from residual beer by *M. purpureus* after 192 h of fermentation. Productivity and specific growth rate values were found as 2.3 UAh$^{-1}$ and 0.03 h$^{-1}$ respectively. Even though the calculated specific growth rate value of *M. purpureus* grown on DM whey was in the range reported by the other authors, pigment productivity was found to be higher than the other substrates. These findings make DM whey a productive and sustainable substrate for the production of red pigment by SHF of *M. purpureus*.

### 3.8. Assesment of Production Cost of Pigment Production

The utilization of agricultural residues and organic waste from agro-industries as carbon sources for the production of value-added products reduce the production cost and provide environmental sustainability. On the other hand, product yield, transporting, handling and pretreatment costs should be considered when wastes are used as inexpensive organic substrates for fermentation processes. The operational cost includes the handling and pretreatment processes of raw materials, fermentation, downstream processes, labor, and maintenance along with the plant and administrative costs. Furthermore, the pretreatment process of the waste substrate should be appropriate in terms of process and environmental sustainability. Physical, chemical and hydrothermal processes are among the several pretreatments applied to agricultural waste materials [58].

As can be seen from Table 1, the amount of pigment produced from soybean, coconut and bagasse is low, therefore transportation and handling costs for utilizing large volumes of waste materials for pigment production will be very high. The concentration of pigment

produced from corn meal supplemented with glucose (6%) and whey (1%) is higher than pigment concentrations when other waste materials are used. The addition of whey to the fermentation medium may have provided the necessary nitrogen and mineral sources for cell proliferation. Since coconut and bagasse contain glucose sources in the form of cellulose, the pigment yield will also be low unless the substrate is pretreated. Hence, the use of cellulosic agricultural waste in fermentation requires a costly pretreatment process and more waste will be generated after the fermentation limiting the environmental sustainability of the process. However, instead of lignocellulosic waste, the use of agro industrial wastes such as whey that does not require many pre-treatment unit operations prior to the fermentation seems to be a more economical and sustainable alternative for the red pigment production process.

**Table 1.** Pigment yields and necessary waste material for red pigment production.

| Agricultural Waste Material | Pigment Yield (mg/g Dry Substrate) | Waste Material (g Dry Substrate/g Pigment) | Cost ($/kg) | References |
|---|---|---|---|---|
| Soy bean residues (Supp. with 6% glucose) | 1.65 | 606.06 | | [2] |
| Coconut residues (Supp. with 6% glucose) | 5.65 | 176.99 | | [2] |
| Bagasse (Supp. with 6% glucose) | 7.5 | 133.33 | | [2] |
| Corn meal (Supp. with 6% glucose) | 20.86 | 47.94 | | [2] |
| Corn meal (Supp. with 6% glucose + 1% whey) | 47.42 | 21.09 | | [2] |
| Demineralized whey (6% glucose equivalent) | 133.3 | 7.50 | 14.92 | This Study |
| Glucose (6%) | 166.67 | 6.00 | 14.84 | Theoretical |

In this study, the operational cost of red pigment production by *M. purpureus* is calculated as 14.92 dollars/kg when commercially processed, demineralized whey was used for pigment production. This calculated cost is close to the pigment production process cost when glucose is used as the carbon source, since both are processed products. Pigment production cost can be reduced by the use of raw liquid whey coming directly from cheese production, but the membrane filtration process needed to concentrate the sugar will possibly increase the production cost of the process. Therefore, it is important to concentrate more on the economic feasibility of the process for future work.

### 4. Conclusions

Suitable substrates rich in nutrients are necessary for microbial growth in the fermentation processes to produce economical, sustainable high value-added bioproducts. In this study, whey from the dairy industry was investigated as a fermentation medium for red pigment production by *M. purpureus*. To optimize fermentation conditions and maximize the yield, the effects of whey type, fermentation methods, initial lactose concentration, MSG concentration, initial pH, and inoculation ratio on red pigment synthesis by *M. purpureus* were systematically investigated. The results showed that DM whey is a suitable substrate for the red pigment synthesis by *M. purpureus* and SHF was highly affected by the tested process parameters. This research provides new insights into the utilization of whey produced in massive amounts and presents a possible solution for serious environmental pollution problems. The highest pigment synthesis was obtained with treated samples of whey (DM and DPDM) in this study. Since the nutrients or inhibitory compounds are different in each form of whey, they might have different effects on biomass formation and product synthesis. Hence, future studies will focus on the full characterization of all whey types and the positive and negative effects of the major and minor constituents on pigment synthesis by *Monascus purpureus*.

**Author Contributions:** D.M.: Investigation, Methodology, Formal Analysis, Writing—Original Draft Preparation, N.A.P.: Calculation, Review & Editing, Y.G.: Investigation, Conceptualization, Resources, Supervision. All authors have read and agreed to the published version of the manuscript.

**Funding:** No funding received for this study.

**Institutional Review Board Statement:** Not applicable.

**Informed Consent Statement:** Not applicable.

**Data Availability Statement:** Not applicable.

**Acknowledgments:** The authors would like to thank Saisamorn Lumyong from Chiang Mai University, Department of Biology, Thailand for kindly supplying the *Monascus purpureus* CMU 001 strain.

**Conflicts of Interest:** The authors declare no conflict of interest.

## Abbreviations

| | |
|---|---|
| W | Whey powder |
| DM | Demineralized whey powder |
| DP | Deproteinized whey powder |
| DPDM | Treated deproteinized demineralized whey powder |
| DPW | Treated deproteinized raw whey powder |
| MSG | Monosodium glutamate |
| PHSF | Prehydrolysis and separate fermentation |

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
