# Peer review of "Utilization of Whey for Red Pigment Production by Monascus purpureus in Submerged Fermentation"

_fermentation, doi:10.3390/fermentation7020075_

Round 1

Reviewer 1 Report

The present study has the objective of producing pigments by Monascus purpureus using the whey as substrate for the fermentation process. The study lacks of some important analyses. In addition, is really important to enrich  the section of Results and Discussion with other observations and comparative analyses.

Lacks the statistic analysis for all figures presented in the results; report is needed if they are or not significative.

The Figure 4 is  not  visible in the pdf file and therefore the relative discussion cannot be evaluated.

To make intriguiting the paper, a comparative analysis must be made. In particular, the advantages to utilize the Monascus purpureus vs Monascus ruber for Red Pigment Production in the Whey  is needed to describe. Also, if present, disadvantages  must be reported. To this purpose, other references are necessary in the manuscript.

Author Response

First of all, we thank you for all the comments and corrections in our manuscript and for your positive effort for evaluating our manuscript. Below you can find the corrections done and comments about our manuscript. 

  1. The present study has the objective of producing pigments by Monascus purpureus using the whey as substrate for the fermentation process. The study lacks of some important analyses. In addition, is really important to enrich the section of Results and Discussion with other observations and comparative analyses.

Response: Thanks for the valuable critics of the reviewer. The composition of whey samples and the effect of major and minor constituents of whey samples on pigment production is lacking in this manuscript. However, as stated in the “Conclusion” part of our paper, “future studies will focus on full characterization of all whey types and the positive and negative effects of the major and minor constituents on pigment synthesis by Monascus purpureus “ (page 12-13). Also in our previous paper (Atalay et al.,2020) on pigment production from residual beer using the same strain, Monascus purpureus CMU001, scanning electron photomicrographs showing the outer morphology of Monascus pigment powder were given.

  1. Lacks the statistic analysis for all figures presented in the results; report is needed if they are or not significative.

Response: We thank the reviewer for correcting our mistake. The statistical method was added to “Method“ section of the manuscript and the results of the statistical analysis are given in all figures by using letters a,b,c as stated below. All the figures (except Figure 7 which is not a bar graph) were reproduced with the results of Duncan’s test.

The below sentences were added to the manuscript on page 4.

“All statistical analyses were performed using the software IBM SPSS (Version 22). All obtained data were analyzed by one way analysis of variance, and tests of significant differences were determined by using Duncan’s test at ‘p < 0.05’. In all the figures, mean values for all the factors given in x-axis that are not followed by the same letter (a-g) are significantly different (p<0.05).”

We also added the following sentence to page 5:

“Similar pigment yield value of 20.2 UA510nm was obtained with deproteinized and demineralized whey powder medium (DPDM). No significant difference (p>0.05) in pigment yields were observed with DM and DPDM media.”

  1. The Figure 4 is not visible in the pdf file and therefore the relative discussion cannot be evaluated.

Response: Figure 4 will be present in the revised version of the manuscript.

  1. To make intriguiting the paper, a comparative analysis must be made. In particular, the advantages to utilize the Monascus purpureusvs Monascusruber for Red Pigment Production in the Whey  is needed to describe. Also, if present, disadvantages  must be reported. To this purpose, other references are necessary in the manuscript.

Response: We thank the reviewer for showing us another point of view in pigment production. Most strains in the genus Monascus belong to four species: M. purpureus, M. pilosus, M.anka and M. ruber and the majority of microorganism for producing pigments and other secondary metabolites is M. purpureus. The aim of the present study is to investigate whey as an alternative low-cost sustainable substrate in the fermentation of Monascus purpureus CMU 001 strain to produce red color pigment for the food industry. In Monascus research some scientists use M.purpureus while some others use M.ruber. We obtained our strain from Chiang Mai University of Thailand and we choosed to study this strain as stated in various parts of the manuscript. To the best of our knowledge, no previous study in the literature has systematically compared the advantages and disadvantages of M. purpureus and M. ruber strains in pigment production.

Reviewer 2 Report

The work seems interesting and the authors approached the topic quite nicely. It seems to me that after minor corrections, or basically slight corrections, I would find the work acceptable.
Below are my minor comments:
1. More precisely in the abstract, I would highlight the purpose of the work.
2. In the introduction, I would focus more precisely on the topic of work and cite the literature more precisely. It looks pretty meager for now, but it's easy to fix.
3. The drawings are subject to a slight correction, so far they are somehow disproportionate in terms of descriptions of the axes to the drawing.
4. Complete the references.

Author Response

First of all, we thank you for all the comments and corrections in our manuscript and for your positive effort for evaluating our manuscript. Below you can find the corrections done and comments about our manuscript. 

The work seems interesting and the authors approached the topic quite nicely. It seems to me that after minor corrections, or basically slight corrections, I would find the work acceptable.

Below are my minor comments:

  1. More precisely in the abstract, I would highlight the purpose of the work.

Response: Thanks for the valuable contribution of the reviewer. The purpose of the work was highlighted in the abstract as stated below (page 1).

“The purpose of this study is to investigate the effect of different fermentation parameters in simultaneous hydrolysis and fermentation (SHF) of whey for pigment production by using Monascus purpureus. The submerged culture fermentation parameters optimized were type and pretreatment of whey, pH, inoculation ratio, substrate concentration, monosodium glutamate (MSG) concentration.”

  1. In the introduction, I would focus more precisely on the topic of work and cite the literature more precisely. It looks pretty meager for now, but it's easy to fix.

Response: The reasons of using pigments and the advantages of microbial pigments (page 1), the properties and use of Monascus pigments (lines 83-96), the importance of utilizing wastes in bioprocesses (page 2), various food industry wastes used to produce Monascus pigments in the literature  (page 2), the properties, composition and the reasons of using  whey in this study (page 2) and finally the aim of this study (pages 2 and 3) were emphasized in the “Introduction” part of the manuscript. Thirty six papers were used in total in the Introduction part.  The Introduction part of our manuscript places the study in a broad context and highlights its importance with a detailed definition of the purpose of the work and its significance.

  1. The drawings are subject to a slight correction, so far they are somehow disproportionate in terms of descriptions of the axes to the drawing.

Response: The drawings are checked once more for corrections and reproduced with the results of Duncan’s test.

  1. Complete the references.

Response: Sixty references were used throughout the manuscript and all of them were cross checked once more in reference list and manuscript. The references were reorganized according to the journal template.

Reviewer 3 Report

is it a paper about TRUE utilization of whey as a substrate or a paper of red pigments production from Monascus purpureus grown on lactose and MSG?

huge levels of lactose (up to 100g/L) and MSG (up to 30g/L)

and then, no whey in Figure 7?

Whey powder (W), demineralized whey powder (DM), deproteinized whey powder  (DP), treated deproteinized raw whey powder (DPW), and treated deproteinized demineralized whey powder (DPDM) were used as fermentation susbtrates.

in Figure 1, where is your control without whey (at all, any form)?

production with MSG and lactose only

fig 1   biomass production is quite the same in all the tested 'whey'

DM, DP, W, DPW, DPDM

red pigment range from 5 to 20 (max x 4)

please give details about the compounds involved in this x4 effect

(detailed composition of each colored extract)

what are the chemical structures of the pigments? (LC-MS-ToF, NMR etc)

the use of whey with Monascus strains is not new

what about the production of lovastatin and citrinin in your conditions?

ref 37 is not suitable (not in English and published in an international peer reviewed journal)

37. Şılbır S. (2019) Bira Atığından Monascus Renk Pigmentleri Üretimi ve Stabilitesinin Belirlenmesi (Doctoral disserta-547 tion, Ege University) Retrieved from YÖK Tez Merkezi, Publication no:590713.

figure 4 is missing in the PDF

what kind of experimental evidence do you provide about true degradation of whey powder (W), demineralized whey powder (DM), deproteinized whey powder  (DP), treated deproteinized raw whey powder (DPW), and treated deproteinized demineralized whey powder (DPDM) by your Monascus strain?

table 1

how do you convert your red pigments AU510nm in mg/g DS?

Author Response

First of all, we thank you for all the comments and corrections in our manuscript and for your positive effort for evaluating our manuscript. Below you can find the corrections done and comments about our manuscript. 

Is it a paper about TRUE utilization of whey as a substrate or a paper of red pigments production from Monascus purpureus grown on lactose and MSG?

Response: The paper investigates the effect of different fermentation parameters in simultaneous hydrolysis and fermentation (SHF) of whey for pigment production by using Monascus purpureus. As lactose is the abundant sugar in whey samples, total sugar analysis were done in terms of lactose and the results were given as lactose concentration (there is lactose in whey). MSG was used as the nitrogen source to increase pigment synthesis as stated by other researchers.

huge levels of lactose (up to 100g/L) and MSG (up to 30g/L)

Response: The demineralized whey powder was diluted to contain the desired lactose concentration and initial lactose concentrations of 25, 50, 75, 100 g/l and MSG concentrations of 2.5, 5.0, 7.5, 10.0, 12.5, 15.0, 20.0, 25.0, 30.0 g/L were tested for pigment production.

and then, no whey in Figure 7?

Response: Whey was used as the fermentation medium in Figure 7 since the subject of the paper was pigment production by using whey. Whey was diluted with distilled water to contain the desired initial lactose concentration. Initial sugar concentration in the manuscript was given in terms of lactose, since lactose is the main sugar in whey samples. But in order to prevent misunderstanding; (demineralized whey diluted to contain 75 g/l lactose, 5 g/l MSG concentrations, ……)” was added to page 10.

Whey powder (W), demineralized whey powder (DM), deproteinized whey powder  (DP), treated deproteinized raw whey powder (DPW), and treated deproteinized demineralized whey powder (DPDM) were used as fermentation susbtrates.

in Figure 1, where is your control without whey (at all, any form)?

Response: The object of the study is to produce red pigments from whey samples. So it was not necessary to produce pigments by using a synthetic medium in which lactose was the only carbon source.

production with MSG and lactose only

Response: We did not produce red pigments by using lactose, but we used lactose in whey for pigment production.  MSG was added as the nitrogen source to increase pigment synthesis as stated also by other authors.

fig 1   biomass production is quite the same in all the tested 'whey'

Response: Yes, biomass concentration was similar in all whey samples. This was explained on page 5 of the manuscript as stated below:

“The biomass concentration values observed in all types of whey are very close to each other and no significant correlation was observed between pigment and biomass production values (Figure 1). This indicates that biomass can reproduce in all types of whey, but the conditions for pigment production can be achieved at low anion and cation concentrations.”

DM, DP, W, DPW, DPDM

red pigment range from 5 to 20 (max x 4)

please give details about the compounds involved in this x4 effect

Response: The higher pigment production values obtained with demineralized (DM) and demineralized and deproteinized whey (DPDM) samples might originate from the removal of different cations and anions (that might have inhibitor effects on pigment synthesis) present in whey samples. This was clearly described on page 5 by citing the relevant literature as stated below:

“Lower pigment synthesis values observed in DP and PDW samples might be originated from the inhibitory effect of high concentration of cations (Na+, NH4+, K+, Mg+2, and Ca+2) and anions (Cl-, SO4-2, PO4-3, and citrate) present in whey medium [43,44]. Since, demineralization process decreased monovalent, divalent cation and anion levels in whey, pigment production values obtained from DM and DPDM are satisfactory compared to the literature [45].”

(detailed composition of each colored extract)

what are the chemical structures of the pigments? (LC-MS-ToF, NMR etc)

Response: Thanks to the valuable contribution of the reviewer. Detailed structural and compositional analysis of the produced pigment will be the subject of our future experiments. As also stated in Conclusion part of the manuscript (below), the effect of major and minor constituents of whey samples will be the subject of another research.

“Since the nutrients or inhibitory compounds are different in each form of whey, they might have different effects on biomass formation and product synthesis. Hence, future studies will focus on full characterization of all whey types and the positive and negative effects of the major and minor constituents on pigment synthesis by Monascus purpureus.”

the use of whey with Monascus strains is not new

what about the production of lovastatin and citrinin in your conditions?

ref 37 is not suitable (not in English and published in an international peer reviewed journal)

  1. Şılbır S. (2019) Bira Atığından Monascus Renk Pigmentleri Üretimi ve Stabilitesinin Belirlenmesi (Doctoral dissertation, Ege University) Retrieved from YÖK Tez Merkezi, Publication no:590713.

Response: The Ph.D. thesis [37] has showed that the strain we used for pigment synthesis did not produce the mycotoxin citrinin, which is a secondary metabolite of Monascus species.

Reference 37 given in the reference list is an open access Ph.D. thesis that can be retrieved from the internet site of the Council of Higher Education of Turkey (Publication 590713). The internet site that can be used to download the Ph.D. thesis is:

https://tez.yok.gov.tr/UlusalTezMerkezi/giris.jsp

The name of the Ph.D. thesis should be written in the search item of the internet site.

figure 4 is missing in the PDF

Response: This will be corrected in the revised version of the manuscript.

what kind of experimental evidence do you provide about true degradation of whey powder (W), demineralized whey powder (DM), deproteinized whey powder  (DP), treated deproteinized raw whey powder (DPW), and treated deproteinized demineralized whey powder (DPDM) by your Monascus strain?

Response: As seen in Figure 1, Monascus purpureus produced biomass in all whey samples.  It is obvious that the hydrolyzed lactose was degraded in the simultaneous hydrolysis and fermentation process and was used as a carbon source by the organism. However pigment synthesis varied among the whey types which showed us that the composition of the whey sample influenced pigment synthesis.

table 1

how do you convert your red pigments AU510nm in mg/g DS?

Response: These two terms were not converted since they are completely different terms. AU510nm is the universal term used in the literature for calculating pigment production and this value is obtained by spectrophotometric readings at 510 nm. On the other hand, “mg/g DS” is milligrams of pigment obtained per gram of used substrate (whey in our study). In other words, it gives us the milligrams of pigment obtained by using 1 gram of whey. The dry pigment in milligrams is obtained by the lyophilization of pigment solution obtained after the fermentation process.

Reviewer 4 Report

The authors in the manuscript "Utilization of Whey for Red Pigment Production by Monascus purpureus in Submerged Fermentation" described the experiments to optimizes and increase the color yield using a fungus and dairy waste. The introduction was well conducted, the information was clear. The methods and the results sections were well described; however, some minor comments need should be improved and address.

In general.

They need to describe and give information about the whey used. In the dairy industry whey can be classified into acid whey or sweet whey-based on its processing conditions. The composition is different, and the first point is the pH (see https://doi.org/10.3168/jds.2020-19038).

The authors need to include statistical analysis. The comparison between samples and treatment needs to be included and the significant difference.

I recommend to organizes the treatments in a table, to be easier for the readers. Even you can include some components and conditions.

The authors need to check through the manuscript some scientific words like h instead hours, min instead minutes, also they need to homogenize the units like g/l or g/L. Even, the taxonomic name used needs to be consistent (e.g. lane 157). The first time they need to include the complete name, after that just the abbreviation.

All the figures need to be checked. E.g Figure 4 does not show the bars. They should also check the units.

To detail.

Lane 19-20. Include the meaning of C, MSG, and N.

L46 and L65. Write the abbreviated name M. purpureus

L70. Define what type of whey

L81-86. I don't find meaning in this sentence. Please delete it.

L130. Explain the demineralization process. Doesn’t change the whey composition after this process?

L132. After autoclave and pH adjusted process, the whey proteins can coagulate. Did the authors see this effect?

L165. Reorganize this sentence.

L180-182. Include the statistical analysis, software, and type of analysis.

L200. I do not agree with this statement. DPDM showed almost the same values. Is it significantly different?

L223. What does "6substrate" mean?

L248. Check M. Ruber

L254. Which is the initial pH? And the pH did not affect the protein coagulation?

Figure 4. The bars in the figure do not appear

L360. Change UA500nm for UA500nm

Figure 5. Change Biyomass for Biomass.

Figure 7. The figure is amazing; however, I recommend using a special Data Analysis and Graphing Software (Origin Lab, SigmaPlot, etc) to show the real changes, e.g pH. Also, the double-axis is not clear. What about the SD?

Table 1. Could the authors address an estimated cost for the rest of the agricultural waste material?

Conclusion section. The author needs to be more concise. I recommend reorganizing this section, addressing the best conditions to produce the red pigments.

Author Response

First of all, we thank you for all the comments and corrections in our manuscript and for your positive effort for evaluating our manuscript. Below you can find the corrections done and comments about our manuscript. 

They need to describe and give information about the whey used. In the dairy industry whey can be classified into acid whey or sweet whey-based on its processing conditions. The composition is different, and the first point is the pH (see https://doi.org/10.3168/jds.2020-19038).

Response: Thanks to the reviewer, this is a good point. “Raw whey powder, demineralized whey powder and deproteinized whey powder were supplied by Enka Süt ve Gıda Mamulleri Sanayi ve Ticaret A.S. located in Konya city, Turkey.” as stated in page 3. All the whey samples were obtained in powder form. Deproteinized and demineralized whey was obtained by the procedure described on page 4. As stated in conclusion part of the manuscript, the composition of each whey sample and the effects of major and minor constituents of each sample on pigment production will be the subject of our future research.

However the specifications for whey powder samples obtained from the supplier are as follows:

Raw whey: sweet whey type, water, 4%; protein, max. 7%; oil content, 1.5 %; minerals 9%; acidity, 0.16% (in lactic acid); lactose, min. 80%; salt, 8%, pH 6.0 (for 10 % solution).

Deproteinized whey: water, 4%; protein, 1-6%; oil content, 1.5 %; minerals 9%; acidity, 0.16% (in lactic acid); lactose, min. 75%; salt, 8%, pH 6.0 (for 10 % solution).

Demineralized whey : Since we obtained the highest pigment production with this substrate, the composition of this whey sample was determined by us and given on page 6 as : “Demineralised whey powder with the highest pigment yield had the following composition (%):  total sugar 70.46± 0.24, protein 5.32±0.05, ash 3.32±0.36, total nitrogen 0.86±0.01, moisture 2.72±0.3.”

We did not declare the above specifications in the manuscript since they are the declared specifications by the supplier, but as we mentioned before major and minor constituents of whey samples and their effect on pigment production will be the subject of our future work.

However, the following sentence was included on page 3 to declare that the samples were sweet whey based samples. “All the whey samples were sweet whey based samples as declared by the supplier.”

The authors need to include statistical analysis. The comparison between samples and treatment needs to be included and the significant difference.

Response: We thank the reviewer for correcting our mistake. The statistical method was added to “Method“ section of the manuscript and the results of the statistical analysis are given in all figures by using letters a,b,c as stated below. All the figures (except Figure 7 which is not a bar graph) were reproduced with the results of Duncan’s test.

The below sentences were added to the manuscript on page 4.

“All statistical analyses were performed using the software IBM SPSS (Version 22). All obtained data were analyzed by one way analysis of variance, and tests of significant differences were determined by using Duncan’s test at ‘p < 0.05’. In all the figures, mean values for all the factors given in x-axis that are not followed by the same letter (a-g) are significantly different (p<0.05).”

We also added the following sentence to page 5:

“Similar pigment yield value of 20.2 UA510nm was obtained with deproteinized and demineralized whey powder medium (DPDM). No significant difference (p>0.05) in pigment yields were observed with DM and DPDM media.”

I recommend to organize the treatments in a table, to be easier for the readers. Even you can include some components and conditions.

Response: We thank the reviewer for giving us this idea. However, we only made deproteinization treatments to whey samples as described on page 3. All the other whey samples in powder form were just diluted with distilled water to contain the desired concentration of lactose.

 The authors need to check through the manuscript some scientific words like h instead hours, min instead minutes, also they need to homogenize the units like g/l or g/L. Even, the taxonomic name used needs to be consistent (e.g. lane 157). The first time they need to include the complete name, after that just the abbreviation.

Response: We thank the reviewer for his/her attention. All the paper was checked once more and the corrections made were highlighted in the manuscript.  

All the figures need to be checked. E.g Figure 4 does not show the bars. They should also check the units.

Response: The units of the figures were checked and g/L were changed as g/l in all the figures. Figure 4 will be present in the revised version of the manuscript.

To detail.

Lane 19-20. Include the meaning of C, MSG, and N.

Response: Corrected and the corrections were highlighted in the Abstract section.

L46 and L65. Write the abbreviated name M. purpureus

Response: Corrected and highlighted in the manuscript.

L70. Define what type of whey

Response: The following sentence was added to page 3 (2.1. Microorganisms and culture media) of the manuscript.

“All the whey samples were sweet whey based samples as declared by the supplier.”

L81-86. I don't find meaning in this sentence. Please delete it.

Response: We could not be sure of the mentioned sentence since the line numbers have changed in the manuscript. We did not take the risk of deleting another sentence that was not mentioned by the reviewer. If the reviewer writes the whole sentence, the editorial might delete it in the publication process.

L130. Explain the demineralization process. Doesn’t change the whey composition after this process?

Response: We only did deproteinization to whey samples as stated on page 4 as “DM and W samples were subjected to deproteinization to obtain DPW and DPDM. “ The details of our deproteinization procedure was given in detail on page 4.

The demineralization process was done by the supplier and the details was not told by the supplier. Demineralization process is expected to decrease the mineral content of whey sample as stated on page 4 as “Since, demineralization process decreased monovalent, divalent cation and anion levels in whey, pigment production values obtained from DM and DPDM are satisfactory compared to the literature [45].”

L132. After autoclave and pH adjusted process, the whey proteins can coagulate. Did the authors see this effect?

Response: The reviewer fingered on a important point. Heat induces protein coagulation in whey. Hence, we also detected a limited protein coagulation in our fermentation substrates with less in deproteinized samples as protein precipitation had already induced in these samples. However, since the pH of the fermentation media was adjusted to 6.0, this protein coagulation was not very evident.

L165. Reorganize this sentence.

Response: The line numbers have changed in the manuscript and we could not be sure of the sentence.

L180-182. Include the statistical analysis, software, and type of analysis.

Response: The statistical method was added to “Method“ section of the manuscript and the results of the statistical analysis are given in all figures by using letters a,b,c as stated below. All the figures (except Figure 7 which is not a bar graph) were reproduced with the results of Duncan’s test.

The below sentences were added to the manuscript on page 4.

“All statistical analyses were performed using the software IBM SPSS (Version 22). All obtained data were analyzed by one way analysis of variance, and tests of significant differences were determined by using Duncan’s test at ‘p < 0.05’. In all the figures, mean values for all the factors given in x-axis that are not followed by the same letter (a-g) are significantly different (p<0.05).”

L200. I do not agree with this statement. DPDM showed almost the same values. Is it significantly different?

Response: Thanks the reviewer for this comment. We added the following sentence to page 5:

“Similar pigment yield value of 20.2 UA510nm was obtained with deproteinized and demineralized whey powder medium (DPDM). No significant difference (p>0.05) in pigment yields were observed with DM and DPDM media.”

We should have choosed a whey sample for further fermentation experiments and we choosed to go on with demineralized whey sample with which slightly higher pigment production value was obtained.

L223. What does "6substrate" mean?

Response: Corrected as “substrate” and highlighted in the manuscript.

L248. Check M. Ruber

Response: Corrected as “M.ruber” (on page 6) and highlighted in the manuscript.

L254. Which is the initial pH? And the pH did not affect the protein coagulation?

Response: The initial pH was adjusted to pH 6.0 (unless otherwise stated) for fermentation trials as stated on page 3. Protein coagulation was more evident in pH trials when pH of fermentation medium was adjusted to 5.0. Lower pH values were not used in fermentation trials.

Figure 4. The bars in the figure do not appear

Response: Corrected in the revised version.

L360. Change UA500nm for UA500nm

Response: Corrected and highlighted in the revised version of the manuscript.

Figure 5. Change Biyomass for Biomass.

Response:  “Biyomass” was changed as “Biomass” in Figure 5.

Figure 7. The figure is amazing; however, I recommend using a special Data Analysis and Graphing Software (Origin Lab, SigmaPlot, etc) to show the real changes, e.g pH. Also, the double-axis is not clear. What about the SD?

Response: Thanks for the positive comments of the reviewer. “The standard deviation of each experimental point ranged from 1.5 to 4.8” sentence was added and highlighted to the caption of Figure 7. Thanks to the reviewer that Data Analysis software will be considered in future publications.

Table 1. Could the authors address an estimated cost for the rest of the agricultural waste material?

Response: The data for the rest of the agricultural waste materials were derived from literature and it will be very hard for us to make a cost analysis for those wastes since we did not work on those wastes and we did not know the exact flowsheet and process conditions for the utilization of these wastes. 

Conclusion section. The author needs to be more concise. I recommend reorganizing this section, addressing the best conditions to produce the red pigments.

Response: The best conditions of pigment production were stated in the Abstract part. Hence we did not mention the same conditions in conclusion section. We preferred to concentrate on the impact and critics of the research in conclusion part.

Round 2

Reviewer 1 Report

The paper has been revised and can be accepted in present form

Regards 

Reviewer 2 Report

The authors very good improvements in this article. 
Hence I still will be supporting the publication of this manuscript. 
For me, this manuscript in its present form is a very good piece of paper. 
I believe that in near future this work will have a lot of interest from readers and also a lot of citations. 
I propose acceptance of this article in this form. 

Reviewer 3 Report

Round 1 with 3 reviewers. I was the only one to recommend to reject.

MS was a little bit improved + majority among reviewers ---> accept